# Epidemiological and Molecular Study of *Cryptosporidium* in Preweaned Calves in Kuwait

**DOI:** 10.3390/ani12141805

**Published:** 2022-07-14

**Authors:** Qais A. H. Majeed, Maha S. AlAzemi, Mohammed T. Al-Sayegh, Nadra-Elwgoud M. I. Abdou

**Affiliations:** 1Department Science, College of Basic Education, PAAET, Aridyia, Farwanyia 23167, Kuwait; ms.alazemi1@paaet.edu.kw (M.S.A.); mt.alsayegh@paaet.edu.kw (M.T.A.-S.); 2Early Warning Center for Transboundary Animal Diseases-Gulf Cooperation Council, PAAFR, 1307 Safat, Rabyia, Farwanyia 21422, Kuwait; 3Department of Medicine and Infectious Diseases, Faculty of Veterinary Medicine, Cairo University, Giza 12211, Egypt

**Keywords:** *Cryptosporidium* spp., *C. parvum* IIaA15G2R1, preweaned calves, risk factors, Kuwait

## Abstract

**Simple Summary:**

Cryptosporidiosis is a global, zoonotic disease of concern. *Cryptosporidium* spp. can infect susceptible hosts via a main fecal–oral route due to cross-contamination of raw food and surface water from reservoir animals in the neighborhood, farms, or slaughter houses, besides some mechanical vectors, such as cockroaches and flies. *Cryptosporidium*
*parvum* alleles are the most common species infecting children, and its potential reservoir is cattle. Hence, understanding the epidemiology of *Cryptosporidium* spp. in preweaned calves, along with the diagnosis of the predominant species and subtypes infecting them, can play a role in preventing *Cryptosporidium* spp. spread in the environment. In this study, *Cryptosporidium*
*parvum* subtype IIaA15G2R1 was the most dominant *Cryptosporidium* spp. detected in preweaned calves in Kuwait. This subtype was recorded previously in Kuwaiti children suffering from diarrhea. Maintaining good personal hygiene in humans and reducing, controlling, or eliminating the causal risk factors in preweaned calves is a superb strategy for preventing and narrowing the spread of this disease.

**Abstract:**

*Cryptosporidium* is a worldwide enteric protozoan parasite that causes gastrointestinal infection in animals, including humans. The most notable species is *Cryptosporidium parvum* because of its zoonotic importance; it is also the leading cause of cryptosporidiosis in preweaned calves. A cross-sectional study was conducted to determine the prevalence of *Cryptosporidium* infection, investigate the potential risk factors, and use molecular diagnosis to identify the predominant *Cryptosporidium* spp. in preweaned calves in Kuwait. Of 175 preweaned calves, *Cryptosporidium* antigens were detected in 58 (33.1%) using rapid lateral immunochromatography assay (IC). Calves less than one month of age (OR = 4.32, *p* = 0.0001) and poor hygiene (OR = 2.85, *p* = 0.0075) were identified as significant risk factors associated with *Cryptosporidium* infection. Molecular identification revealed that *C. parvum* (62.8%) was the dominant species infecting preweaned calves in Kuwait. In contrast, *C. bovis* and *C. andersoni* were recorded at 5.7% and 2.9%, respectively. All *C. parvum* gp60 nucleotide sequences were subtype IIaA15G2R1. Calves could be a source of *C. parvum* infection due to the similarity of the subtypes recorded previously in Kuwaiti children and preweaned calves in this study. Therefore, more research is needed to understand the *Cryptosporidium* transmission cycle in Kuwait.

## 1. Introduction

*Cryptosporidium* is a globally distributed protozoan parasite. It accounts for most cryptosporidiosis cases in newborn farm animals, and is a significant causative agent of diarrhea in children in many parts of the world [1,2,3].

*C. parvum* is one of the main enteropathogens of neonatal calf diarrhea; it infects preweaned calves ≤ six weeks of age [4]. However, calves ranging from one to three weeks old appear to be the most susceptible age. Infected calves suffer from acute profuse diarrhea, high morbidity, possible mortality, and reduced growth rate as long-term effects of cryptosporidiosis [3]. This species is also predominant in humans in the Middle East, notably its IIa allele, implying that cattle may be involved in zoonotic infections [5,6,7].

Kuwait is a small country in a desert region. Vegetation is highly scarce given the type of climate and soil. The climate is continental, with a dry, hot season (April–November) and a mild, cold, wet season (December–March). Dust storms are expected during the hot season, and temperatures can reach 50 °C. Due to this harsh climate and limited vegetation, dairy farming presents unique challenges. The 2020 census recorded 31,484 cattle in the country [8]. Cattle are reared for milk production under an intensive farming system with zero grazing. The breed of most dairy cattle is Friesian, and their farms are confined to the Sulaibiya area. Most dairy cattle were imported from Germany and Holland to rebuild the dairy sector after the destruction of livestock during the Iraq invasion.

No studies have been conducted in Kuwait to determine the prevalence of *Cryptosporidium*, identify *Cryptosporidium* spp. and subtypes in cattle, or investigate their public health significance. In contrast, many research papers have been published on the molecular characterization of bovine *Cryptosporidium* spp. and their global prevalence [1,6,7,9,10,11,12,13,14,15]. Consequently, the objectives of this study were to estimate the prevalence and determine the risk factors of *Cryptosporidium* infection in preweaned calves (≤3 months of age) and identify the genotypes and subtypes of *Cryptosporidium* in this animal species, furthermore to assess their public health importance in Kuwait.

## 2. Materials and Methods

### 2.1. Study Design, Data, and Sample Collection

Between October 2014 and September 2015, a cross-sectional study was conducted. A single visit was made to each farm that took part in the study to collect samples and data. For data collection from each farm, a structured questionnaire (open-ended, closed-ended, dichotomous, or multiple choice) was created. The host factors (breed, age, sex), the environmental factors (location, season) were the subjects of the data collection. In addition, management factors, such as management system, herd size, frequency of cleaning and bedding change, presence of feed and water troughs, source of water, separation of age groups, etc., were collected

All cattle farms in Kuwait are private dairy farms of Friesian breed, located in Sulaibiya (29°28′56.0″ N, 47°81′80.0″ E), and reared under an intensive farming system with zero grazing. The farms were supplied with desalinated potable water from a municipal source. The presence of feed and water troughs, maternity facilities, and the separation of age groups were almost identical among all the farms. Data on the frequency of cleaning and bedding changes were consistent with on-farm cleanliness visual monitoring during the visit to determine the hygienic farm status as poor or good. The farms were chosen without knowing whether or not they were infected with *Cryptosporidium*.

Twenty-two dairy cattle farms were visited in the Sulaibiya area. The overall number of dairy cattle on farms visited was 9365 (882 preweaned calves). Herd size of cattle ranged from 12 to 2400 animals (median 300). The Epi Info 7-Stat Cal tool was used to determine the sample size, and systematic random sampling was used to select animals from each visited farm. One hundred seventy-five preweaned calves (≤3 months of age) were randomly selected to examine their fecal samples. Using a sterile screw-capped bottle labelled with the animal’s data (such as sex, age, and health status) and the date of the sample, 5–10 gm of feces were obtained directly from the rectum or shortly after defecation. The specimens were stored at 4 °C or processed within 48 h after being placed in an icebox and transported to the lab.

### 2.2. Processing of Samples

Each sample was divided into two portions in the laboratory: the first part for detecting *Cryptosporidium* antigen by IC. The second portion was used for storage either in 2.5% potassium dichromate or at −20 °C and was sent to Prof. Dr. Lihua Xiao (College of Veterinary Medicine, South China Agricultural University, Guangdong Province, China) for molecular diagnosis and typing of *Cryptosporidium* spp. if the sample was diagnosed as positive for *Cryptosporidium* antigen by rapid lateral immunochromatographic assay (IC).

### 2.3. Detection of Cryptosporidium in Fecal Samples

A commercial rapid lateral immunochromatographic assay (Anigen Rapid BoviD-4 Ag Test Kit; BioNote Inc., Gyeonggi-do, Korea) was used to detect *Cryptosporidium*, rotavirus, coronavirus, and *E. coli* K99 antigens. The procedures and results’ interpretation were performed following the manufacturer’s prescripts.

### 2.4. DNA Extraction, PCR Amplification, and Subtyping

For the typing and subtyping of *Cryptosporidium* spp., 35 IC specimens, that tested positive for the parasite, were chosen. Prior to DNA isolation, the specimens, stored in potassium dichromate, were centrifuged double in DH2O. Using the FastDNA SPIN kit for soil (MP Biomedicals, Santa Ana, CA, USA), DNA was pulled from all samples. Afterward, nested PCR, targeting an approximately 830-bp region of the small subunit (SSU) rRNA gene, was used to check the recovered DNA for *Cryptosporidium* species [16,17].

By performing a restriction fragment length polymorphism (RFLP) analysis on the secondary PCR products with *Ssp*I and *Mbo*II, as previously published [17], *Cryptosporidium* spp. were distinguished. Using the negative control (reagent water) and the positive control (*Cryptosporidium baileyi* DNA), each sample was examined at least twice. The secondary PCR results from typical specimens were sequence analyzed to validate the identification of *Cryptosporidium* species.

Using an ABI 3130 genetic analyzer, all gp60 gene PCR products and representative SSU rRNA gene PCR products from *C. parvum* were sequenced (Applied Biosystems, Foster City, CA, USA). In order to identify *Cryptosporidium* spp. (based on SSU rRNA sequences) and subtypes, for *C. parvum* subtypes, bi-directional sequences were obtained and assembled using the ChromasPro (version 1.5) software (Technelysium Pty Ltd, South Brisbane, Australia. Webpage: http://technelysium.com.au/?page_id=27 accessed on 25 April 2018). These sequences were then aligned with each other, and referenced sequences of each gene were downloaded from Gen Bank using ClustalX (Conway Institute UCD, Dublin, Republic of Ireland. Webpage: http://www.Clustal.Org/, accessed on 30 April 2018). The existing *Cryptosporidium* subtype nomenclature approach was used to name the *C. parvum* subtypes [5].

### 2.5. Statistical Analysis

The Microsoft Office program’s MS Access^®^ database information system (Microsoft Corporation, Redmond, WA, USA) was used to store test results and questionnaires. The data was exported to SXW^®^ for statistical analysis (Statistix 10, Analytical software, Tallahassee, FL, USA). The independent variables under study were age group, sex, season, herd size, and farm hygiene status. The correlation between variables and the prevalence of *Cryptosporidium*-infection in calves was examined using univariate analysis (Chi-square test, χ^2^) at a 95% Confidence Interval. The significant variables (*p* ≤ 0.05) were analyzed using multivariate stepwise logistic regression. The Hosmer–Lemeshow test was applied to determine the goodness of fit for the logistic regression model, and *p* > 0.05 indicated a good fit model.

## 3. Results

The prevalence of *Cryptosporidium* among the examined calves was 33% (58/175). Rotavirus, coronavirus, and *E. coli* were detected in 16 (9%), 3 (2%), and 35 (20%) of the samples, respectively (Figure 1). *Cryptosporidium* spp. mono-infection was discovered in 42 preweaned calves (72%). Co-infections with *Cryptosporidium* were 14% (8/58) for rotavirus, 10% (6/58) for *E. coli*, and 4% (2/58) for rotavirus and *E. coli* (Figure 1).

The results of the univariate analysis showed that four variables were distinguished as putative risk variables for *Cryptosporidium* infection in the examined calves (Table 1). There was a significant distinction in *Cryptosporidium* prevalence rates among the various age group of the examined calves (*p* = 0.0000), with the highest prevalence in calves less than one month of age (50.6%; CI 39.4–61.7). The *Cryptosporidium* infection rate was higher in the wet than in the dry season (37.6%; CI 29.1–46.7; *p* = 0.0476). Similarly, herd size significantly influenced the *Cryptosporidium* infection rate (*p* = 0.0062). The prevalence rate in calves reared in large herd sizes (42.4%; CI 32.2–53.1) was higher than those in small (22.9%; CI 14.4–33.4). Furthermore, farm hygiene status significantly affected the rate of infection (*p* = 0.0005). Calves reared on farms with poor farm hygiene (42.9%; CI 33.5–53.0) had more infections than those with good hygiene (17.6%; CI 9.5–28.8). On the other hand, the sex of the examined calves did not significantly affect the *Cryptosporidium* prevalence rate (*p* = 0.7412); hence, this factor was excluded from the multivariable analysis. The four variables (age group, season, herd size, and farm hygiene) were subjected to multivariate logistic regression, which revealed that calves less than one month of age (OR = 4.32; CI 2.09–8.57; *p* = 0.0001) and poor hygiene (OR = 2.85; CI 1.32–6.13; *p* = 0.0075) were the significant risk factors identified in this study, as shown in Table 2.

Molecular identification of the positive fecal samples showed that *Cryptosporidium* was detected in 71.4% (25/35). The *Cryptosporidium* species identified by RFLP analysis of the SSU rRNA PCR product included *C. parvum* in 22 (62.8%), *C. bovis* in 2 (5.7%), and *C. andersoni* in 1 (2.9%) of the examined samples (Figure 2). Subtyping of *C. parvum* at the gp60 locus revealed that all 22 samples (100%) belonged to one subtype IIaA15G2R1 (Figure 2).

## 4. Discussion

Bovine cryptosporidiosis is globally distributed and has been reported as a considerable risk factor for calf enteritis [3,4,18,19]. This study was the first attempt to identify the risk factors and molecular typing of species involved in *Cryptosporidium* infection among preweaned calves in Kuwait. Many epidemiological studies have been conducted to detect cryptosporidiosis in calves worldwide [10,15,20,21,22,23,24].

In the present study, the overall infection rate of *Cryptosporidium* in the examined calves was 33.1% (95 % CI 26.2–40.6%). In the Middle East, previous studies of *Cryptosporidium* infection in preweaned calves reported prevalence rates of 47.9% in Iraq [25], 18.7% in Jordan [10], 14.7% in Iran [20], 84% in Algeria [13], 58.3% in Sudan [14], 9.7% in Egypt [15], and 53.6% in Turkey [23]. These variations in infection rates could be due to geographical differences in the prevalence of *Cryptosporidium* infections, besides other factors related to differences in diagnostic methods, sampling strategies, farm management, and hygiene [3]. For instance, the rate of infection may be elevated if specimens were only collected from diarrheic preweaned calves. Additionally, prevalence rate could be underestimated in a cross-sectional study compared with a longitudinal study, because the shedding profile of *Cryptosporidium* oocysts is intermittent and could be missed [26]. However, cross-sectional studies can provide more consistent evaluations of disease risk factors than longitudinal studies [27].

The present study used the rapid IC assay to diagnose cryptosporidiosis in preweaned calves. The IC assay’s advantages are accurate, rapid, cost-effective, and easy to conduct, as it does not require other specialized instruments compared with microscopy, ELISA, and PCR techniques. Many investigators study the diagnostic performance of these assays. Sensitivity rates for different commercially available IC detecting *Cryptosporidium* copro-antigen have previously been reported to vary from 75% to 100%, whereas their specificity rates ranged from 92% to 100% [28,29,30].

The present study observed a significant correlation between *Cryptosporidium* infection and calves’ age. Calves less than one month of age were more likely to harbor *Cryptosporidium*, and infection rates diminished progressively with age. These results were consistent with most existing studies, showing that this protozoan is most common in neonatal calves [3,4,22,23]. The highest peak of *Cryptosporidium* infection rate was reported in two-week-old calves [23,31]. This could be due to their immature immune systems [32].

In the present study, multivariate analysis identified farm hygiene status as a risk factor for *Cryptosporidium* infection in preweaned calves. The prevalence of *Cryptosporidium* is approximately three times higher in farms with poor hygienic status (OR = 2.85; 95% CI = 1.32–6.13) compared to those with good hygiene. Similar results were previously observed [24,33,34]. This result could be explained by the fact that a dirty and muddy farm could create appropriate climatic conditions for the presence or survival of *Cryptosporidium* oocysts in the farms or animal houses. Consequently, it raises the risk of calves contracting *Cryptosporidium* infection by contaminating their feed and water [35].

In this study, sex, season, and herd size was not considered risk factors for *Cryptosporidium* infection in preweaned calves in Kuwait. Similar results have been observed. The absence of association between season and the presence of the infection has been formerly described, as season is not considered a risk factor for cryptosporidiosis in countries where the temperature never drops below the freezing point [36,37,38]. At the same time, previous studies found that the sex of the animal and the herd size did not affect the prevalence of *Cryptosporidium* infection [24,34,39].

*Cryptosporidium parvum* was the predominant species, representing 62.9% of the identified species by RFLP, followed by *C. bovis* at 5.7% and *C. andersoni* at 2.9%. Previous studies reported the predominance of *C. parvum* in preweaned calves globally [11,15,40,41,42,43,44]. However, *C. andersoni* is mainly reported in postweaned and adults, more than in preweaned calves. It infects the abomasum and causes maldigestion, weight loss, and a decrease in milk yield [45,46,47]. In this study, *C. andersoni* was identified in one preweaned calf. A similar result was obtained previously; *C. andersoni* was detected in 1% (one 4-week-old calf/161) of *Cryptosporidium* spp. isolated from preweaned calves in the USA [40]. Kváč et al. [41] reported that *C. andersoni* was identified in 13% (21/161) of dairy calves of up to two months of age, suggesting that direct contact with adult animals could be a risk factor for *C. andersoni* infection. Although most researchers declared that *C. bovis* are found predominantly in postweaned calves and never reported clinical disease in different age groups of infected cattle [3], *C. bovis* oocyst shedding was observed in a five-day old calf [38].

In the present study, all *C. parvum* isolates were subtyped as IIaA15G2R1. Similarly, in some Middle Eastern countries, subtype IIa was the only reported subtype family with different alleles identified from cattle [10,13]. Subtypes IIa and IId were found in other countries, with the subtype IIa being more dominant [6,12,43], whereas in Egypt, IId was the predominant subtype over IIa [48,49]. Subtype IId alleles were the only reported subtype from diarrheic calves in Sudan [14].

In Kuwait, a previous study reported that the subtype IId alleles were the most commonly identified in small ruminants [50]. Similar results were recorded in Romania, where most of the examined cattle (86.7%) were infected with subtype IIa, while the ovine isolates belonged to subtype IId [51]. Furthermore, a previous study in Spain recorded that 98% of *C. parvum* subtypes from small ruminants belonged to subtype IId, suggesting that this subtype is adapted to small ruminants [52]. In contrast, subtype IIa has been reported in cattle worldwide [1]. The distribution of the two subtypes is most likely related to animal management, rather than host adaptation. Cattle and small ruminants are kept in separate areas in Kuwait; thus, cross-transmission is difficult due to this separation. In Turkey, cattle herd mingling during grazing increased the opportunity for *Cryptosporidium* transmission among herds, resulting in higher genetic diversity and the appearance of many genotypically mixed isolates in animal populations [12]. Additionally, animal movement, such as transportation or importation, may form a parasite population structure in an infected animal population in a given area [53,54].

*Cryptosporidium parvum* subtype IIaA15G2R1, reported in cattle in this study, is also the predominant allele in cattle in European countries [55]. Thus, the source of *Cryptosporidium* genotypes and *C. parvum* subtypes in Kuwait may be Germany and Netherlands, from which the cattle were imported to rebuild livestock after destruction during the Iraqi invasion.

Regarding zoonotic importance, only two studies previously reported *C. parvum* subtypes in children in Kuwait. Iqbal et al. [56] identified *C. parvum* IIa as the dominant subtype of *Cryptosporidium* in diarrheic children. Simultaneously, Sulaiman et al. [5] recorded the *C. parvum* subtype IIaA15G2R1 as the predominant *Cryptosporidium* spp. in children in Kuwait. *Cryptosporidium parvum* infects a broad variety of mammals, and is a significant zoonotic diseases issue [57]. More research is required to fully understand the *C. parvum* transmission cycle in Kuwait, because its predominance in Kuwaiti children suggests an animal to human transfer, particularly when subtyping findings are taken into account.

## 5. Conclusions

In conclusion, calf age and farm hygiene were the most significant risk factors for *Cryptosporidiosis* in preweaned calves in Kuwait. Calves less than one month of age were more likely to harbor *Cryptosporidium*, and infection rates reduced progressively with age. *Cryptosporidium parvum* was the most commonly identified species from preweaned calves less than three months of age, preceded by *C. bovis* and *C. andersoni*. All *C. parvum* gp60 nucleotide sequences were subtype IIaA15G2R1. Since the distribution of *Cryptosporidium parvum* subtypes in children and cattle is comparable, these animals could be a source of *Cryptosporidium infection* in Kuwaiti humans. Further study is needed to clarify *the Cryptosporidium* transmission cycle in Kuwait.

## Figures and Tables

**Figure 1 animals-12-01805-f001:**
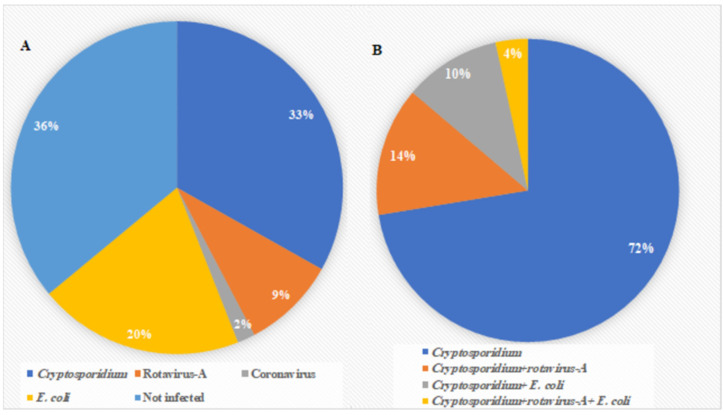
Results of rapid IC assay: (**A**) Prevalence of the four pathogens (*Cryptosporidium,* rotavirus-A, coronavirus, and *E. coli*) detected in 175 preweaned calves. (**B**) Prevalence of *Cryptosporidium* mono-infection and co-infections with other pathogens detected by IC.

**Figure 2 animals-12-01805-f002:**
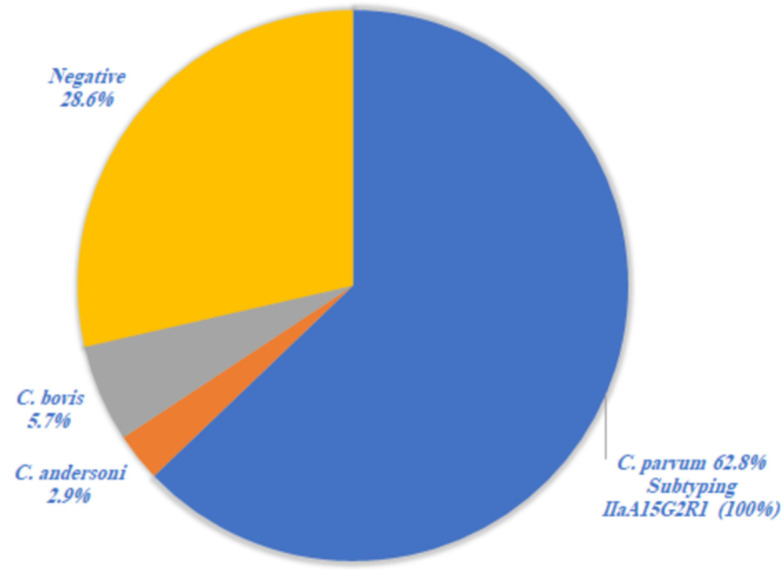
*Cryptosporidium* spp. identified by RFLP analysis in positive IC samples (No. 35) and subtyping of *C. parvum* at gp60 locus.

**Table 1 animals-12-01805-t001:** Univariate analysis results of variables associated with *Cryptosporidium*-infected calves (No. 175) in Kuwait.

Risk Factors		No. of Samples	Prevalence of *Cryptosporidium*		*p*-Value
			No.	% (95% CI) *	
**Age group** **(in months)**	**<1**	83	42	50.6 (39.4–61.7)	0.0000
	**≥1–** **<2**	43	9	20.9 (10.0–36.0)	
	**≥2–** **≤3**	49	7	14.3 (5.9–27.2)	
**Sex**	**Male**	95	33	34.7 (25.2–45.2)	0.7412
	**Female**	80	25	31.3 (21.4–42.6)	
**Season**	**Wet**	125	47	37.6 (29.1–46.7)	0.0476
	**Dry**	50	11	22.0 (11.5–36.0)	
**Herd Size** **(heads)**	**Large > 300**	92	39	42.4 (32.2–53.1)	0.0062
	**Small ≤** **300**	83	19	22.9 (14.4–33.4)	
**Farm hygiene**	**Good**	68	12	17.7 (9.5–28.8)	0.0005
	**Poor**	107	46	42.9 (33.5–53.0)	
**Overall prevalence**		175	58	33.1 (26.2–40.6)	

* CI Confidence Interval.

**Table 2 animals-12-01805-t002:** Multivariate stepwise logistic regression analysis * of risk factors for *Cryptosporidium* infection that were significant using univariate analysis.

Variables	Coefficient β	Std. Error	*p*-Value	OR **	95% CI ***	
					Lower	Upper
**Age group (in months) < 1**	1.442	0.360	0.0001	4.23	2.09	8.57
**Poor hygiene**	1.046	0.391	0.0075	2.85	1.32	6.13

* *p* value (Hosmer–Lemeshow goodness of fit test) = 0.1315, ** OR odds ratio, *** CI Confidence Interval.

## Data Availability

The datasets used and/or analyzed during the current study are not publicly available, but are available from the corresponding author on reasonable request.

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
