# Peer review of "Epidemiological and Molecular Study of Cryptosporidium in Preweaned Calves in Kuwait"

_animals, 2022, doi:10.3390/ani12141805_

Round 1
Reviewer 1 Report
Review
Article: Epidemiological and molecular study of Cryptosporidium in preweaned calves in Kuwait
Introduction:
Introduction on the epidemiology of Cryptosporidium is clear. The purpose of the study is well described
Methods:
Lines 89 to 98: Please clarify about the minimum sample size calculated by EpiInfo and percentage of sample losses. In the systematic sampling, was each farm considered a conglomerate? Complex sampling? Please clarify if sample design has been incorporated into estimates.
Lines 134 to 141: Please clarify how the authors worked with missing data in variables. Change the p_value on line 140 to 0.05. How did the authors perform the logistic regression? Stepwise or forward? The goodness of fit should be described in the text.
Results:
Please, insert the confidence interval 95% in all prevalences and OR
Table 1: please correct the < and > symbols
Table 2: insert footnote about goodness of fit
Author Response
We extend our thanks to you for your efforts and valuable suggestions that improved our manuscript.
Point by point response
Response to reviewers’ comments
Methods: Lines 89 to 98: Please clarify about the minimum sample size calculated by EpiInfo and percentage of sample losses. In the systematic sampling, was each farm considered a conglomerate? Complex sampling? Please clarify if sample design has been incorporated into estimates.
We calculate sample size using Stat Calc. tool of Epi Info 7, first step we select the type of study design (the study design had been incorporated into the estimates by selecting the cross-sectional tab). In order to calculate sample size using Epi Info 7, one requires to provide confidence level which is usually set at 95% so percentage of sample losses would be 5%. Increase the confidence level and/or the power (usually set at 80 %) increase the samples size, so the minimum sample size calculated by Epi Info depends on the factors added. Regarding systematic sampling, each farm was considered as a one-unit and not complex sampling. I rewrote this paragraph to clarify it.
Lines 134 to 141: Please clarify how the authors worked with missing data in variables. Change the p_value on line 140 to 0.05. How did the authors perform the logistic regression? Stepwise or forward? The goodness of fit should be described in the text.
In this study, there was no missed data in the study variables; but in previous studies, I had missed data in some variables and we left the cell empty in the file of the statistic program.
The p value on line 140 corrected to 0.05.
We used stepwise logistic regression
The Hosmer-Lemeshow test was applied to determine the goodness of fit for the logistic regression model and p value > 0.05 indicated a good fit model.
Results: Please, insert the confidence interval 95% in all prevalences and OR
Inserted in the table and text
Table 1: please correct the < and > symbols Corrected
Table 2: insert footnote about goodness of fit
Hosmer-Lemeshow goodness of fit test p = 0.1315

Reviewer 2 Report
Profound study. I guess You mixed up something with “<” and “>”. Please check it. Other comments are minor.
|
Line |
Comment |
|
32 |
It is not clear what is the risk, if You read only the abstract. Is it compared to younger calvesor contact to calves of age > 1mt.? |
|
118 |
2.3. and 2.4. shows that You were using 2 different test, cow-side test (IC and PRC) witj verymuch different sensitivity ans specifity. When You select only IC positive samples for PCRYou meight have a bias by false negatives samples or straines with a weak reaction to IC under the detection level of a certain strain which would not be detected by IC? This would influence the measured prevalence. This has to be discussed and elucidated. |
|
170 |
You have to check the classes/groups: |
|
170 |
Same problem here > or < |
|
208 |
see 2.3. and 2.4. |
|
216 |
multi..... |
|
227 |
Season is not a risk faktor for Cryptos in counties where the temperature never drops belowthe freezing point. |
|
267 |
Is it possible that farmworkers at the major dairies in Kuwait can transmit Crypto to and fromthese dairies to their own privat small herd of small ruminants? |
|
286 |
again: check > and ≥ |
Author Response
Thank you sincerely for your efforts and valuable suggestions that improved our manuscript.
Line 32: It is not clear what is the risk, if You read only the abstract. Is it compared to younger calves or contact to calves of age > 1mt.?
It is compared to calves less than one month age.
Line 118: 2.3. and 2.4. shows that You were using 2 different test, cowside test (IC and PRC) witj very much different sensitivity and specifity. When You select only IC positive samples for PCR You meight have a bias by false negatives samples or straines with a weak reaction to IC under the detection level of a certain strain which would not be detected by IC? This would influence the measured prevalence. This has to be discussed and elucidated.
This study was part of a project where we study Cryptosporidium infection in Livestock in Kuwait. We used modified Ziehl-Neelsen method, Sandwich ELISA, and IC. Best results compared to PCR was IC for that reason we chose IC as the diagnostic tool to depend on it. In addition, PCR results could be affected by low-density oocysts in the fecal samples that may contain PCR inhibitors [1]. Also, the preservatives used may penetrate the oocysts, which cannot be removed by washing, consequently inhibiting PCR results, especially in the long-term storage of fecal samples [2,3]. We collected the samples throughout one year study, then we sent them all to Prof. Dr. Lihua Xiao so some samples where stored for more than one year.
We illustrated why we used IC in Discussion (204 - 210) “The present study used the rapid IC assay to diagnose cryptosporidiosis in preweaned calves. The IC assay's advantages are accurate, rapid, cost-effective, and easy to conduct, as it does not require other specialized instruments compared with microscopy, ELISA, and PCR techniques. Many investigators study the diagnostic performance of these assays. Sensitivity rates for different commercially available IC detecting Cryptosporidium copro-antigen have previously been reported to vary from 75% to 100%, whereas their specificity rates ranged from 92% to 100%.”
1- Elwin, K; Robinson, G; Hadfield, SJ; Fairclough, HV; Iturriza-Gómara, M and Chalmers RM (2012). A comparison of two approaches to extracting Cryptosporidium DNA from human stools as measured by a real-time PCR assay. J. Microbiol. Methods, 89(1):38–40.
2- Chalmers, RM and Katzer, F (2013). Looking for Cryptosporidium: the application of advances in detection and diagnosis. Trends Parasitol., 29(5):237–251.
3- Kuk, S, Yaza, S and Centinkaya U (2012). Stool sample storage conditions for the preservation of Giardia intestinalis Mem. Inst. Oswaldo. Cruz., 107(8):965.
Line 170: You have to check the classes/groups: > 1: greater than 1 ≤ 1 - ≥ 2: smaler or equal to 1 until larger or equal to 2 ≤ 2 - ≥ 3: smaler or equal to 2 until larger or equal to 3 Check that in the whole text.
Checked and corrected throughout the whole text.
Line 170: Same problem here > or < y < x: y is smaler than x y > x: y is greater than x If You change that in all Your results it make sense concerning Crypto.
Corrected
Line 208: see 2.3. and 2.4. I rewrote Materials and Methods
Line 216: multi..... Corrected
Line 227: Season is not a risk faktor for Cryptos in counties where the temperature never dr ops belowthe freezing point.
I added a sentence to clarify that
267: Is it possible that farmworkers at the maj or dairies in Kuwait can transmit Crypto t o and from these dairies to their own privat small herd of small ruminants?
Probably yes, but this needs further study including samples from farm workers and small ruminants to illustrate that.
286: again: check > and ≥
Thank you, I’ve checked and corrected them throughout the whole text.

Round 2
Reviewer 2 Report
Thank You for respecting my comments.